# Elimination of Hepatitis C in Southern Italy: A Model of HCV Screening and Linkage to Care among Hospitalized Patients at Different Hospital Divisions

**DOI:** 10.3390/v14051096

**Published:** 2022-05-19

**Authors:** Valerio Rosato, Loreta A. Kondili, Riccardo Nevola, Pasquale Perillo, Davide Mastrocinque, Alessio Aghemo, Ernesto Claar

**Affiliations:** 1Liver Unit, Ospedale Evangelico Betania, 80147 Naples, Italy; valeriorosato@gmail.com (V.R.); riccardo.nevola@unicampania.it (R.N.); pasqualeperillo@hotmail.it (P.P.); davidemastrocinque4@gmail.com (D.M.); 2Center for Global Health, Istituto Superiore di Sanità, 00161 Rome, Italy; loreta.kondili@iss.it; 3Department of Biomedical Sciences, Humanitas University, 20090 Pieve Emanuele, Italy; alessio.aghemo@gmail.com; 4Division of Internal Medicine and Hepatology, Department of Gastroenterology, Humanitas Research Hospital IRCCS, 20089 Rozzano, Italy

**Keywords:** HCV, opportunistic screening, linkage to care, HCV elimination

## Abstract

**Background:** Free-of-charge HCV screening in some key populations and in 1969–1989 birth cohorts has been funded in Italy as the first step to diagnosing individuals who are infected but asymptomatic. The aim of this study is to evaluate the feasibility of an opportunistic HCV screening and its linkage to care. **Methods:** A hospital-based HCV screening was conducted as a routine test for in-patients admitted to the Evangelical Hospital Betania of Naples from January 2020 to May 2021. All consecutive in-patients were screened for the HCV antibody (HCV-Ab) at the time of their admission to the hospital, and those born prior to year 2000 were included in the study. HCV-RNA testing was required for those not previously treated and without antiviral treatment contraindications. For in-patients with an active infection, treatment started soon after hospital admission. **Results:** Among 12,665 inpatients consecutively screened, 510 (4%) were HCV-Ab positive. The HCV-Ab positivity rate increased with age, reaching the highest prevalence (9.49%) in those born before 1947. Among patients positive for HCV, 118 (23.1%) had been previously treated, 172 (33.9%) had been discharged before being tested for HCV-RNA, and 26 (5.1%) had not been tested for short life expectancy. Of 194 (38% of HCV-Ab+) patients who were tested for HCV-RNA, 91 (46.2%) were HCV-RNA positive. Of patients with active infection, 33 (36%) were admitted to the liver unit with signs of liver damage either not previously diagnosed or diagnosed but unlinked to care for HCV infection. Of the patients positive for HCV-RNA, 87 (95.6%) started treatment; all achieved sustained virological response. **Conclusion:** HCV active infection has been frequently found in patients with comorbidities admitted in the hospital in Southern Italy. To achieve HCV elimination in Italy, broader screening strategies are required. In addition to screening of the 1969–1989 birth cohort of individuals unaware of their infection status, diagnosis and linkage to care of patients with known liver damage is strictly required. Hospital screening is feasible, but prompt reflex testing for identifying HCV-active infections is necessary to increase diagnosis and subsequent linkage to care.

## 1. Introduction

Chronic hepatitis C virus (HCV) infection is a serious public-health problem, with an estimated global prevalence of 0.7% (0.7–0.9%), corresponding to approximately 56.8 million viremic HCV infections globally [1]. Italy has been one of the countries with the highest HCV prevalence and death rate for HCV-induced liver disease in Europe [2], as well as the European country with the highest number of patients treated for HCV with Direct Acting Antivirals DAAs. To date more than 230,000 patients have been treated with DAAs, regardless of liver-fibrosis stage [3]. Recent estimates have shown that, without an intensive case finding, the pool of diagnosed and treated patients will run out in Italy from 2023 to 2025, leaving a significant number of infected individuals without diagnosis and cure [4]. The estimation of patients with HCV chronic active infection in Italy consisted (in 2020) of about 280,000 patients, who are potentially asymptomatic, and additionally about 120,000 patients with advanced liver fibrosis/cirrhosis with an uncured HCV infection [5,6]. In particular, the prevalence rates of HCV infection in Southern Italy have been reported to be high, with peaks of up to 8.2% among persons of advanced ages (60–70 years) [7,8]. Despite the fact that universal HCV screening is a cost-effective intervention in Italy, to achieve the HCV elimination goal by 2030, the general-population screening of 1948–1988 birth cohorts and of those with risk factors for HCV infection independent of their age has been recommended [9,10]. Several barriers, the most important one being the impact of the COVID-19 pandemic, have not allowed the full implementation of this program of HCV screening [11,12]. However, to overcome multiple obstacles, a variety of strategies were applied in Italy during the COVID-19 pandemic [13,14]. This study was conducted on patients consecutively admitted to a large referral hospital in Naples, Italy, to evaluate the feasibility of a hospital HCV-opportunistic-screening during the period of COVID-19 pandemic restrictions (January 2020–May 2021). The final goal was to evaluate if hospital opportunistic screening of admitted patients could become a model of efficient screening and linkage to care for HCV infection in highly endemic countries.

## 2. Materials and Methods

An opportunistic hospital screening and referral model was conducted throughout all divisions of the Evangelic Hospital Betania of Naples from January 2020 to May 2021. The presence of HCV antibodies (HCV-Ab) was tested as a routine standard of care in all consecutive patients admitted to the hospital in different divisions (ordinary hospitalization, day hospital, and day surgery), regardless of the reason for being hospitalized.

As a standard-of-care practice, a report of the identified patients positive for HCV-Ab was sent on a regular basis to the Liver Unit specialists, and the indication to confirm the HCV-active infection by HCV-RNA test, when appropriate, was promptly provided to each hospital division for identification of the active infection. The biochemical characteristics and the previous HCV treatment data of HCV-Ab positive patients were evaluated prior to performing the HCV-RNA test. HCV-Ab was identified by immunoassay for qualitative detection (Elecsys^®^ Anti-HCV II, Cobas^®^, Roche, Basel, Switzerland), while the serum HCV RNA was assessed by real time PCR with lower-limit quantitation of 10 IU/mL (GeneXpert^®^, Xpert HCV viral load^®^, Cepheid, Sunnyvale, CA, USA).

Patients previously treated for HCV infection, or not suitable for a DAA treatment during hospital admission due to severe comorbidities and/or with short life expectancy were not further evaluated for confirmation of active infection upon admission. Patients with the diagnosis of active hepatitis C were immediately linked to care to start a DAA treatment. For patients who had more than one admission to Betania hospital during the study period, only the first access was evaluated. Liver-disease staging was performed for all patients who were linked to care, as required by the Italian Drug Medicines Agency (AIFA), in order to start DAA therapy.

**Statistical Analysis**: continuous variables are expressed as mean values and standard deviations, and categorical data were represented as frequencies and percentages. The Student *t*-test and the Mann–Whitney U test were performed to compare differences in the values of continuous variables. Fisher’s exact test and the chi-square test with the Yates correction were used to evaluate the significance of associations among categorical variables. Statistical significance was defined as *p* < 0.05 (two-tailed test, 95% confidence interval). Statistical analyses were performed with SPSS software (SPSS version 20, SPSS Inc., Chicago, IL, USA).

**Ethics**: This study was conducted in accordance with the guidelines of the Declaration of Helsinki and the principles of good clinical practice. The patients’ data were collected and conserved according to the general data-protection regulation (EU) 2016/679. Informed consent was obtained from all subjects involved in the study. The notification for this study was provided to the Ethical Committee of ASL Na 1 Centro as required for retrospective evaluations in the clinical practice (7 February 2022).

## 3. Results

From January 2020 to the end of May 2021, 15,724 consecutive patients, were referred to Betania Hospital for ordinary hospitalization or day hospital or day surgery). Data of all 15,724 patients, equal to 100% of patients hospitalized during the study period, were retrospectively evaluated. Newborn patients (2736 pts) admitted to the neonatal intensive care unit, and all patients born before 2000 (323 pts) who tested HCV-Ab negative were excluded from this analysis. The youngest patient positive for HCV-Ab was 23 years old, and based on this result, the study population of this study included 12,665 adult individuals aged 20 years or older.

### 3.1. Study Population

Among the 12,665 patients screened, 510 (4%) tested HCV-Ab positive. Among patients born after 1948, males had a significantly higher HCV-Ab prevalence compared with females (5.06% versus 3.56% *p* < 0.024). Among patients born before 1947, this difference was not detected, showing instead a slightly higher prevalence among females (8.5 vs. 10.3 *p* = 0.085).

The flow chart of the study population is shown in the Figure 1. Of 510 patients who tested positive by first-level screening (HCV-Ab positivity), 118 (23.01%) were previously treated for HCV infection. For 26 patients (5.1%), it was not feasible to confirm HCV-active infection due to admission for severe clinical presentation mainly in elderly age, whereas 172 patients (33.8% of HCV-Ab positive individuals) were discharged from the hospital prior to the HCV-RNA determination. Of the remaining 194 HCV-Ab patients positive (38%), who were unaware of a potential HCV infection, HCV-RNA testing was performed during their hospital admission. Of them, 91 (46%) tested positive for HCV and were defined as patients with active HCV infection. Eight-seven patients (95.6%) were treated during hospital admission, while four (4.4%) were not evaluated for treatment during the hospital admission due to severe comorbidities and potential short life expectancy. All (100%) treated patients achieved a sustained virological response at week 12 (SVR 12).

The distribution of active infections by HCV-Ab positivity according to the birth cohort is shown in the Figure 2. Overall, HCV-Ab positivity was 4%. It was significantly higher in older ages and decreased significantly in younger ages (birth cohorts after 1968), whereas among the available patients in whom the HCV-RNA confirmation was required, the rate of active infection among patients positive for HCV-Ab was 29.6%. The rate of active infection was not significantly different between the birth cohorts. However, we found a slightly higher rate of HCV RNA in the 1969–1989 birth cohort compared to the 1968–1948 one (33% vs. 22%) and a rate of 33% of active infection in patients older than 80 years.

### 3.2. Rate of Infection among the Hospiral Divisions

The distribution of HCV-active infection in different hospital divisions is shown in Table 1.

Thirty-five (38.4%) of the patients with HCV-RNA positivity were admitted to the liver unit for severe liver disease with unknown etiology until hospital admission. Among patients with an active HCV infection, 61.5% (56 of 91 patients with an active infection) were admitted to other than the liver unit divisions (mainly in intensive care units and orthopedics, but also in surgery, cardiology, gynecology, senology, as well as ophthalmology).

### 3.3. Biochemical Characteristics of Patients with Active Infection

Table 2 shows demographic and biochemical characteristics of patients with an active infection. Among patients diagnosed with an active HCV infection admitted to different hospital divisions, 22 (62.8%) of those admitted to the liver unit; 25 (44.6%) of those admitted to other divisions had altered transaminase levels; and 23 (65.7%) and 16 (28.5%) had an increased Fib 4 > 3.25 in the liver unit and other divisions, respectively, indicating clinical liver damage due to HCV not diagnosed or diagnosed but yet not cured. 

## 4. Discussion

In the present study, we reported the results of an opportunistic screening in in-patients of a big hospital in Naples, a country where HCV infection has been reported to be highly endemic [7].

The in-patients younger than 20 years of age were not affected by HCV infection, a result that confirms several previously reported data. The HCV-Ab prevalence was lower in younger individuals, ranging from 0%, in those younger than 23 years of age, to 1.19%, in the cohort born from 1969 to 1989, and from 4.18%, in those born from 1948 to 1968, to 9.49%, in those born before the year 1948, confirming the decreasing HCV-infection risk in younger populations in Italy, known as the cohort effect [15,16,17]. The HCV Ab prevalence in patients with different comorbidities was similar to the HCV Ab prevalence found in patients with renal disease in Italy, but lower than the prevalence found in patients of 1945–1944 cohort of age, conducted in South of Italy (Apulia region) reflecting differences in the infection burden in various populations and settings in Italy.

Based on the previously recommended HCV screening in Italy, the birth cohort 1948–1968 screening should be promptly guaranteed to achieve the HCV elimination goals. The results of this study indicate the opportunistic hospital screening in patients with various comorbidities as a feasible strategy to diagnose and subsequently easily link to care the infected patients [18,19]

Regarding the distribution of patients with an active infection (HCV RNA positive) in different hospital admission divisions, overall, 38% (35/91) of patients were admitted to a liver unit for severe liver damage not previously diagnosed as HCV-related. The lack of diagnosis in people with severe liver damage has been also documented by the AIFA DAA monitoring registry, which reported more than 20% of DAA treatment in patients with liver cirrhosis since 2019 to date [3]. As has been recently estimated, active screening of individuals with advanced liver disease in Italy will diagnose more than 100,000 patients who need an urgent linkage to care and treatment to eradicate HCV infection in order to stop liver disease progression [5,6]. Altered transaminase levels and Fib 4 > 3.25 were found in about 28% of patients admitted in divisions other than the liver unit and diagnosed during hospital admission with active infection (HCV RNA positive), indicating liver damage despite the different reasons for the hospital admission. Our data clearly shows the need for an increase in awareness among general practitioners and other health care specialists, who take care of patients with different comorbidities, to address the HCV disease control. This is also important because HCV eradication has been shown to significantly improve not only the outcomes of liver disease, but also of several comorbidities such as cryoglobulinemic vasculitis, diabetes, cardiovascular and renal disease, etc. [20,21].

Regarding asymptomatic persons with active HCV infection, they are mainly present in younger age populations with a shorter duration of infection and without comorbidities, thus less representative compared to older patients admitted to the hospital. Data of this study indicate that younger individuals of the general population have a lower prevalence of HCV-Ab, though a high proportion of them (more than 30%) have an active infection (HCV RNA positive). HCV-active screening of people born between 1969 and 1989; people who inject drugs (PWID), followed by addiction services; and inmates, independently of their age, has been funded by the Italian National Health System as the first key step to achieving the HCV-elimination in Italy [10]. Addressing first previous or active PWID, prisoners, as well as other at-risk populations (migrants, sex workers, and people having undergone risky nosocomial or esthetic procedures) [5,6,10,22,23,24] will give the possibility to identify HCV infected people in a sexually active age, often completely asymptomatic and without perception of the infectious risk and reduce the infection burden, limiting the infection spread and the progression to further disease stages [25]. Opportunistic screening for the younger population, is a rapid and efficient way to and cure HCV infection; however, this should not be the only way to reach younger populations, because people belonging to these populations are less likely to have hospital admission 

Data of our study could only confirm that screening could be useful in different hospital divisions but could not give indications of the real prevalence of active infections due to the lack of confirmation of active infection in 33.8% of patients positive for HCV-Ab. In accordance with this data, studies conducted in different contexts have confirmed the significant loss of patients due to the two steps required for confirmation of an HCV-active infection [26,27]. Reflex testing permits the one-step confirmation of the HCV-active infection in patients unaware of their infection status, and it is strongly suggested in hospital settings [27,28]. However, as opposed to what has been reported in this study, a preliminary questionnaire regarding the patient’s awareness of HCV- and/or antiviral-treatment status should be performed prior to HCV reflex testing during the hospital-admission.

## 5. Conclusions

To achieve HCV elimination in Italy, broader screening strategies are required. In addition to screening of the 1969–1989 birth cohort of individuals unaware of their infection status, diagnosis and linkage to care of patients with known liver damage is strictly required. Opportunistic hospital screening is an effective strategy and in-hospital reflex testing of individuals unaware of HCV status can lead to the immediate engagement and cure of patients diagnosed with an active HCV infection. It is of paramount importance to increase the awareness of general practitioners and other health care professionals on the impact of HCV and its role in liver and extrahepatic diseases.

## Figures and Tables

**Figure 1 viruses-14-01096-f001:**
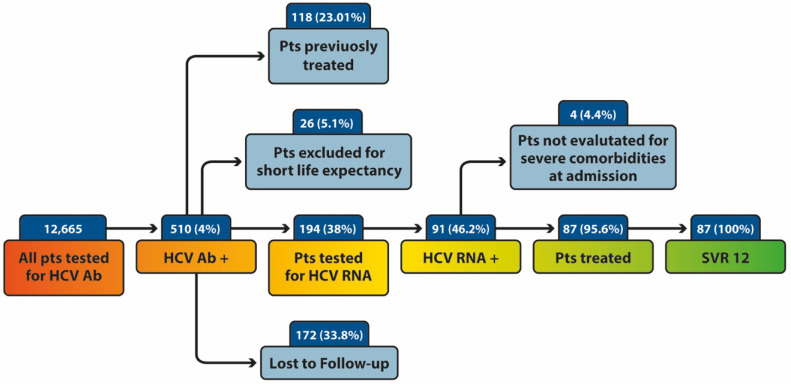
The flow chart of the study population. Pts: patients; SVR: Sustained Virological Response.

**Figure 2 viruses-14-01096-f002:**
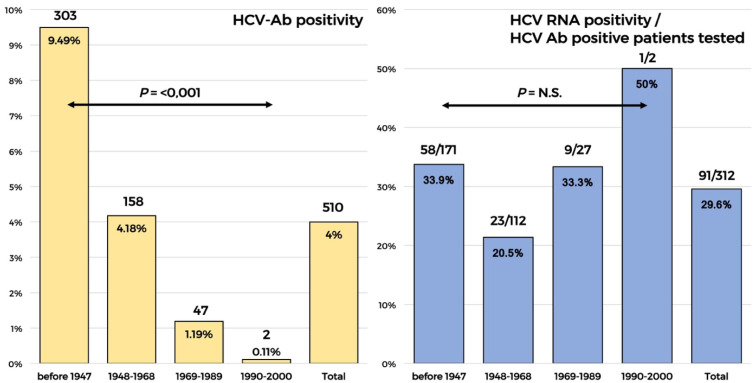
HCV-Ab prevalence and the proportion of active infection detected by birth cohort.

**Table 1 viruses-14-01096-t001:** HCV active infection among patients tested between different hospital divisions.

Age Groups	Cardiology	Gynecology Senology	Liver Unit	ICU	Medicine	Ophthalmology	Orthopedics	Surgery
Beyond 1990	0	0/1 (100)	1/1 (100)	0	0	0	0	0
1989–1969	0	0/3 (0)	1/7 (14.2)	1/2 (50)	2/3 (66.6)	0	2/4 (50)	3/6 (50)
1948–1968	2/7 (28.5)	1/9 (11.1)	10/39 (25.6)	1/1 (100)	2/10 (20)	2/21 (9.5)	4/11 (36.3)	1/14 (7.1)
Before 1947	3/14 (21.4)	3/8 (37.5)	23/52 (44.2)	3/9 (33.3)	6/25 (24)	6/33 (18.1)	7/16 (43.7)	7/15 (46.6)
Total	5/21 (23.8)	4/22 (18.1)	35/99 (35.3)	5/12 (41.6)	10/38 (26.3)	8/54 (14.8)	13/31 (42)	11/35 (31.4)

Data are expressed as *n* HCV-active infection/*n* of patients with HCV-Ab positivity (percentage) ICU: intensive-care unit.

**Table 2 viruses-14-01096-t002:** Biochemical characteristics of patients with active infection admitted to liver unit and to other hospital divisions.

	Other Hospital Divisions	Liver Unit	*p*-Value
*n*	56	35	
Age (year)	72 (±14)	74 (±13)	0.379
male	24 (42.8)	21 (60)	0.184
ALT (U/L)	35 (±34)	153 (±595)	0.385
AST (U/L)	45 (±58)	119 (±300)	0.016
Increased transaminase values *	25 (44.6)	22 (62.8)	0.056
GGT (U/L)	76 (±128)	106 (±119)	0.092
Platelets (103/mL)	182 (±77)	151 (±77)	0.033
Bilirubin (mg/dL)	0.79 (±1.85)	2.29 (±4.47)	0.003
FIB-4 > 3.25	16 (28.5)	23 (65.7)	0.001

Note: Mean (±SD) for continuous variables and *n* (percentage) for categorical variables ALT: Alanine aminotransferase; AST: Aspartate aminotransferase; FIB-4: fibrosis-4 score; GGT: gamma glutamyl transpeptidase; * entailed as a value of AST and/or ALT > 35 U/L.

## Data Availability

Not applicable here. Only aggregated data have been analyzed.

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
