# Peer review of "Elimination of Hepatitis C in Southern Italy: A Model of HCV Screening and Linkage to Care among Hospitalized Patients at Different Hospital Divisions"

_viruses, 2022, doi:10.3390/v14051096_

Round 1
Reviewer 1 Report
It is an interesting study which aims to identify HCV infected individuals when they were first admitted at a hospital in Italy, from January 2020 to May 2021. All admitted patients were screened for HCV antibody HCV-Ab.
The authors conclude that HCV infection is frequently found in patients with comorbidities in Southern Italy and that HCV elimination requires larger than 1969-1989 cohort screening of patients unaware of the infectious status and HCV.
It involves 12.665 inpatients consecutively screened. Overall HCV Ab positivity was 4%. It was significantly higher in older ages and decreased significantly in younger ages
However, I think a few points could be clarified.
- Inclusion criteria is not clear. In the abstract we find: “All consecutive in-patients were screened for HCV antibody (HCV-Ab) at their first hospital admission”. In Methods we read: “HCV antibodies (HCV-Ab) was tested as a routine standard of care in all consecutive patients, aged 20 years or older...”
I think this point should be detailed because great part of the discussion and conclusion is about the age of the cohort. And also, it would be interesting to listen to the author’s opinion on recommendation on screening in infants and adolescents.
- Regarding HCV diagnostic tests, it would be important to detail technical information on the tests which were used.
- Regarding risk factors for HCV infection, it would be important to more information on that. Maybe the authors could comment on that.
The authors mention that in Italy, free of charge HCV screening is offered to key populations and 1969-1989 birth cohorts, as the first step to diagnose asymptomatic individuals.
One of the author’s conclusions, based on data from this manuscript (see above) is that HCV elimination in Italy would require larger than 1969-1989 cohort screening (as currently proposed by medical authority in Italy).
I think it would be plausible to suppose that maybe, among the 510(4%) individuals who were HCV-Ab positive and identified in the study, a great part might be eventually included in high-risk population (previous or active PWID, prisoners, people who received blood transfusion in the past, HIV co-infected, etc). If that is the case, the recommendation of universal screening may not be the most adequate conclusion. I would like to know a few comments on that.
- As one last suggestion , I think the discussion could be a little bit more concise and focused on the results obtained.
Author Response
We thank the reviewer for the helpful comments.
Please see the attachment.

Reviewer 2 Report
The manuscript is well written and comprehensive, with a solid methodology and a good presentation of the results. It contains valuable data from clinical practice. A well-described linkage to care of newly diagnosed patients with HCV infection. The study is in line with the subject of the special issue of the journal. I suggest publishing the paper.
Author Response
We thank the reviewer for the helpful comments. We hope that the last version of our manuscript is suitable to be published in Viruses.
Round 2
Reviewer 1 Report
I believe the manuscript could be accepted in its present form.
No additional comments.